# Advanced Diagnostics of Respiratory Distress Syndrome in Premature Infants Treated with Surfactant and Budesonide through Computer-Assisted Chest X-ray Analysis

**DOI:** 10.3390/diagnostics14020214

**Published:** 2024-01-19

**Authors:** Tijana Prodanovic, Suzana Petrovic Savic, Nikola Prodanovic, Aleksandra Simovic, Suzana Zivojinovic, Jelena Cekovic Djordjevic, Dragana Savic

**Affiliations:** 1Department of Pediatrics, Faculty of Medical Sciences, University of Kragujevac, Svetozara Markovica 69, 34000 Kragujevac, Serbia; tijanaprodanovic86@gmail.com (T.P.); aleksandra.simovic@yahoo.com (A.S.); zivojinovicsuzana@yahoo.com (S.Z.); j.cekovic86@gmail.com (J.C.D.); drsavicdragana@gmail.com (D.S.); 2Center for Neonatology, Pediatric Clinic, University Clinical Center Kragujevac, Zmaj Jovina 30, 34000 Kragujevac, Serbia; 3Department for Production Engineering, Faculty of Engineering, University of Kragujevac, Sestre Janjic 6, 34000 Kragujevac, Serbia; petrovic.suzana@gmail.com; 4Department of Surgery, Faculty of Medical Sciences, University of Kragujevac, Svetozara Markovica 69, 34000 Kragujevac, Serbia; 5Clinic for Orthopaedic and Trauma Surgery, University Clinical Center Kragujevac, Zmaj Jovina 30, 34000 Kragujevac, Serbia

**Keywords:** respiratory distress syndrome, chest X-ray, diagnostics, image segmentation, performance evaluation

## Abstract

This research addresses the respiratory distress syndrome (RDS) in preterm newborns caused by insufficient surfactant synthesis, which can lead to serious complications, including pneumothorax, pulmonary hypertension, and pulmonary hemorrhage, increasing the risk of a fatal outcome. By analyzing chest radiographs and blood gases, we specifically focus on the significant contributions of these parameters to the diagnosis and analysis of the recovery of patients with RDS. The study involved 32 preterm newborns, and the analysis of gas parameters before and after the administration of surfactants and inhalation corticosteroid therapy revealed statistically significant changes in values of parameters such as FiO_2_, pH, pCO_2_, HCO_3_, and BE (Sig. < 0.05), while the pO_2_ parameter showed a potential change (Sig. = 0.061). Parallel to this, the research emphasizes the development of a lung segmentation algorithm implemented in the MATLAB programming environment. The key steps of the algorithm include preprocessing, segmentation, and visualization for a more detailed understanding of the recovery dynamics after RDS. These algorithms have achieved promising results, with a global accuracy of 0.93 ± 0.06, precision of 0.81 ± 0.16, and an F-score of 0.82 ± 0.14. These results highlight the potential application of algorithms in the analysis and monitoring of recovery in newborns with RDS, also underscoring the need for further development of software solutions in medicine, particularly in neonatology, to enhance the diagnosis and treatment of preterm newborns with respiratory distress syndrome.

## 1. Introduction

Prematurely born infants, due to the immaturity of their organ systems, are exposed to an increased risk of morbidity and mortality. Respiratory distress syndrome (RDS) is characterized by alveolar collapse and the development of microatelectasis due to a lack of surfactant. The frequency of this condition decreases as gestational age increases; all infants born between the 22nd and 24th weeks of gestation will have respiratory distress syndrome, while the incidence in infants with a body weight between 1250 and 1500 g will be 25% [1]. RDS is considered the main cause of pneumothorax, pulmonary hypertension, and pulmonary hemorrhage in newborns. These complications increase the likelihood of a fatal outcome as well as the development of bronchopulmonary dysplasia in surviving newborns. Timely diagnosis and an appropriate therapeutic approach play a crucial role in improving the prognosis of respiratory distress syndrome [2,3].

The clinical presentation of RDS is characterized by tachypnea, cyanosis, flaring of the nostrils, intercostal retractions, subcostal retractions, and expiratory grunting. The diagnosis of RDS is established based on typical anamnestic data, clinical findings, gas analysis, and chest radiography. Chest radiography is often considered the “gold standard” for diagnosis. The X-ray may reveal the presence of diffuse microatelectasis and characteristic changes indicating inadequately ventilated pulmonary alveoli. Depending on the degree of progression, radiographic findings of RDS can be classified into five grades, as follows [3,4,5]:Grade I—Bilateral lung fields are generally non-translucent due to reduced aeration. On the X-ray, diffuse fine granulations caused by collapsed alveoli are visible, as is the presence of a reticular shadow, which is a result of air in the bronchioles;Grade II—In addition to the characteristics of the first stage, there are also patchy and streaky opacities;Grade III—Opacity of lung parenchyma significantly increases. In addition to fine-grained, patchy opacities, there is also a loss of clear heart boundaries;Grade IV—Lung tissue appears hazy with a bronchogram over the cardiac shadow;Grade V—Complete lung fields are covered with diffuse opacification (clinically manifesting as “white lungs”), and the heart border is not visible on the X-ray.

Radiography, as an essential diagnostic technique, plays a crucial role in medicine, enabling precise visualization of internal body structures. The efficiency of radiography can be significantly enhanced by applying sophisticated image processing techniques, with a particular focus on the segmentation of X-ray images. Segmentation, as a key step in the analysis of radiological data, allows for the accurate isolation of regions of interest, such as the lungs, for clearer and more detailed insights. This process is essential for improving the accuracy of diagnosis and timely treatment, especially in situations where anatomical variations, pathological changes, or other irregularities may complicate traditional analysis methods [6,7,8,9,10].

With the development of neonatal intensive care units, the survival of more premature and critically ill newborns has increased. Consequently, the frequency of complications such as respiratory distress syndrome and, among them, bronchopulmonary dysplasia, one of the most severe complications of preterm birth, has also risen. To prevent the most severe respiratory complication of respiratory distress syndrome, bronchopulmonary dysplasia, systemic corticosteroid therapy has been applied, which has had a positive effect on lung function but has also been associated with numerous systemic complications. Long-term follow-up studies indicate that the administration of high doses of dexamethasone in the early neonatal period results in poorer neurodevelopmental outcomes [11]. In order to induce favorable effects of corticosteroid therapy on the lungs while minimizing systemic side effects, corticosteroids are increasingly administered locally. This is performed through instillation with surfactant or through inhalation [12,13].

In the literature, a variety of algorithms for chest X-ray image segmentation and lung boundary extraction can be found. Therefore, algorithms can be classified into the following basic categories: (1) rule-based, (2) pixel classification-based, (3) model-based, hybrid, and (4) deep learning-based algorithms [6].

Algorithms in the rule-based category employ sequential steps and heuristic assumptions to identify the lung region. They heavily rely on the concept of a level set, which integrates global statistics, prior shape, and edge information. The initial level set is initiated through a mask calculated using rule-based steps such as thresholding, morphological operations, and connected component analysis [7]. Yang X. et al. chose to segment the lungs using such an approach on CT images. They initially employed a threshold for pre-segmentation to automatically select the starting point, facilitating the application of region growth. Subsequently, they applied morphological post-processing to enhance the segmentation effects [8]. For early detection of breast cancer, a similar methodology was applied by Mehmood M. et al. They isolated tumor regions through a segmentation process using thresholding. Additionally, they improved the segmentation results by applying morphological operations. Since these algorithms demonstrated effectiveness as a basis for image classification, they subsequently applied machine learning techniques to enhance the speed and efficiency of the classification process [9].

In the domain of pixel classification, algorithms rely on the analysis of low-level visual features and prior shapes. An active shape model, as a paradigmatic representative of this approach, shapes its representation through the distribution of landmark points on training images. This model is then adapted to a test image through fine-tuning of distribution parameters, making it relevant in the context of visual data analysis [10]. Hybrid approaches to segmentation integrate the advantages of different algorithms to create an integrated method capable of overcoming challenges in lung detection [14]. In their study, Medeiros A.G. et al. applied an innovative fast morphological geodesic active contour method for lung segmentation. To achieve precise segmentation, they use an initial contour that approximates the shape of the lungs and then adapts it to the actual lung shape after defining the lung boundaries. To further enhance lung segmentation, they apply morphological operations [15]. Vijh S et al. introduced an innovative approach for early detection, diagnosis, and prediction aimed at improving patient treatment and preventive measures. Their methodology emphasizes the segmentation of the region of interest based on a global threshold and the application of morphological operations to achieve a more precise representation of segmented lungs. They further optimized desired features by combining two metaheuristic algorithms, specifically the Whale Optimization Algorithm (WOA) and the Adaptive Particle Swarm Optimization (APSO) algorithm. The classification was performed using a convolutional neural network (CNN) classification technique. [16]. These results also confirm the effectiveness of deep learning-based algorithms, which are trained on large datasets and achieve the most accurate results compared to previous approaches [17]. Notably, Demiroğlu U et al. underscored the importance of utilizing deep learning in predicting and diagnosing lung cancer from CT scans, aiming to mitigate potential human errors. For this purpose, they utilized pre-trained models, DarkNet-53 and DenseNet-201, to develop a highly accurate hybrid classification method [18].

Applied algorithms for lung detection are commonly used on X-ray images of adults with an unchanged appearance of lung anatomy. However, in cases of pathology or altered lung anatomy, this can affect the intensity distribution in lung regions and result in unclear lung outlines, posing a challenge for segmentation algorithms. Besides the lack of algorithms for detecting pathologically altered lung regions, research on pediatric X-ray images of the chest is limited in the literature. The appearance of lungs in children, especially in newborns, deviates from the appearance in adults due to the ongoing process of incomplete alveolarization [7].

Given the current relevance of the topic and intensive research in the field of RDS, as well as the potential for improving existing diagnostic procedures, the focus is directed with particular attention to the analysis of chest X-rays and gas analysis. This approach aims to investigate the key contributions of these parameters in the diagnosis of RDS, providing a deeper insight into the complexity of the disease and opening space for innovations in medical approaches.

## 2. Materials and Methods

### 2.1. Patients

Out of a total of 117 premature newborns who were hospitalized in the Neonatal Intensive Care Unit of the Neonatology Center at the Clinic of Pediatrics, University Clinical Center in Kragujevac, during the period from 1 February 2022 to 1 December 2022, the study included 32 infants who met the inclusion criteria. Their gestational age ranged from 25 to 36 weeks. Birth weight varied from 770 g to 3250 g. The diagnosis of RDS was established based on clinical indicators, X-ray images, and gas analysis of arterialized capillary blood.

Parents of prematurely born children were informed about the examination procedure in accordance with the rules of the Declaration of Helsinki and Good Clinical Practice. The study had the approval of the local Ethics Committee (Approval Number 01/22/3/1 from 11 January 2022), and they voluntarily consented to the examination.

The criteria for including premature newborns in the study were as follows:Premature newborns born before the completion of the 37th gestational week;Premature newborns requiring the administration of exogenous surfactant and invasive mechanical ventilation within the first 6 h of life;Premature newborns requiring a second dose of surfactant by the end of the second day of life at the latest;X-ray images are classified as IV or V grade according to the BomseII classification;Parents have signed informed consent to participate in the study.

The criteria excluding premature newborns from the study were as follows:Premature newborns requiring systemic corticosteroid therapy according to the protocol;Premature newborns with congenital anomalies;Premature newborns require the administration of exogenous surfactant and non-invasive respiratory support;Premature newborns require only one dose of surfactant.

All patients included in the study were administered inhalations of 0.25 mg/kg/12 h Budesonide from the third day of life for a duration of 14 days.

For all cases participating in the study, the following parameters were monitored:Maternal history: age, parity, pregnancies, previous miscarriages, stillbirths, neonatal deaths, as well as acute and/or chronic illnesses;Newborn’s anthropometric measures at birth: body weight, body length, head circumference, chest circumference, and Apgar score;Laboratory analyses: complete blood count, C-reactive protein, glucose levels, gas analysis (pH-pCO_2_-pO_2_-HCO_3_-BE), and electrolyte levels (Na+, K+, Ca++) in arterialized capillary blood;Radiographic images: chest X-ray images were monitored.

### 2.2. Statistical Analysis

For the purpose of a detailed analysis of the results in newborns with RDS, descriptive statistics were employed to provide insight into key parameters. This analysis serves as a foundation for further interpretation and discussion of the results.

Additionally, a comparative analysis of gas measurements before and after the administration of surfactant and inhaled corticosteroids was conducted using a paired samples *t*-test. This statistical approach enables the identification of significant changes in respiratory parameters following surfactant and inhaled corticosteroid therapy. The paired samples *t*-test was chosen because of its ability to detect statistically significant differences between related measurements, in this case, measurements before and after the administration of surfactant and inhaled corticosteroids.

### 2.3. X-ray Image Processing

In an effort to better understand the recovery dynamics after RDS, this study focuses on the development and application of lung segmentation algorithms. The goal is to provide a more precise insight into structural changes in the lungs during the recovery phases.

All chest radiographic images were collected at the University Clinical Center Kragujevac to ensure the consistency and accuracy of the results. The X-ray images were in .jpeg format with a resolution of 1024 × 1024 pixels. The images used were two-dimensional chest images in the anteroposterior projection. For each patient, we collected three images, namely the following:IX-ray image: within the first 6 h after the newborn’s birth, before the first dose of surfactant;IIX-ray image: on the second day of life before starting corticosteroid therapy;IIIX-ray image: on the 16th day of life after the administration of inhaled corticosteroid therapy.

The processing of X-ray images was conducted at the Center for Integrated Product and Process Development at the Faculty of Engineering, University of Kragujevac. For this purpose, the MATLAB programming environment “https://www.mathworks.com/ (accessed 15 November 2023)” was utilized. An algorithm for the segmentation of X-ray images was developed, consisting of several key image processing steps, including preprocessing, segmentation, and visualization. The lung segmentation algorithm is illustrated in Figure 1.

The segmentation algorithm comprised several key steps. First, binary conversion was performed using an appropriate threshold, transforming pixel values into a binary representation. To address potential gaps in segmented regions, a dilation procedure was applied to strengthen the contours of objects. Subsequently, smaller objects were removed to retain larger, more relevant segments. The obtained binary image underwent lung segmentation, and a mask was used to isolate the pulmonary region. Closing operations were then applied to enhance the definition of lung regions, ensuring better representation.

Starting from the image loaded from the corresponding database, it is displayed on the screen and treated mathematically (Figure 2). The image is denoted as [7,19]:(1)$$X\in R^{\left(MxN\right)},\;where\;X\left(i,\;j\right)\in \left[0,\;255\right]\;for\;i=1,\;2,\ldots ,\;M\;and\;j=1,\;2,\;\ldots ,\;N,$$
where
*R*—the set of real numbers;*M*—the number of rows in the image;*N*—the number of columns in the image; *X*(*i*, *j*)—the pixel intensity at position (*i*, *j*).

After this step, the image *X*(*i*,*j*) undergoes further processing by converting it into a binary image *Y*(*i*,*j*) through the application of an appropriate threshold *T* (Figure 3). This process enables the transformation of complex image data into binary values, where pixel values above the threshold become white pixels (1), while values below the threshold become black pixels (0). This conversion facilitates further analysis and extraction of relevant information from the image. The mathematical expression describing this procedure is as follows [7,9,19]:(2)$$Y\left(i,j\right)=\left\{\begin{array}{cc} 1, & X\left(i,j\right)>T\\ 0, & X\left(i,j\right)\leq T \end{array}\right.,$$
where
*T*—the threshold;*X*(*i*, *j*)—the pixel intensity at position (*i*, *j*) in the original X-ray image;*Y*(*i*, *j*)—the pixel value at position (*i*, *j*) in the binary image after applying the threshold *T*.

In the segmentation process, there may be the formation of white regions that are not completely closed or contain unwanted gaps. To address this issue, we apply the dilation procedure (Figure 4). Dilation can be mathematically represented as the convolution of the binary image *Y* with a structural element *B*, resulting in a new binary image *Z*. This step plays a crucial role in strengthening the contours of objects, smoothing their edges, and eliminating small interruptions within regions. The mathematical expression defining dilation is written as follows [7,10,16]:(3)Z=Y⊕B,
where
*Z*—the resulting binary image after applying dilation;*Y*—the original binary image;*B*—the structural element used for dilation;⊕—the convolution operation (or opening) between image *Y* and the structural element *B*.

Removing smaller objects from the binary image aims to clean the image from minor noise, unwanted artifacts, and other small objects that may be present [10]. This phase is essential as it retains only larger segments or regions, which often represent the main objects of interest in the image, while smaller and irrelevant objects are eliminated (Figure 5). This ensures the preservation of essential information, reduces structural complexity, and enhances accuracy.

A binary matrix *Z* is defined to represent the original image with dimensions *M* × *N*, where *Z*(*i*,*j*) can have a value of 0 (representing a black pixel) or 1 (representing a white pixel). The set of regions in the image will be denoted as *S*, where each region *R*(*i*) in *S* has its area (number of pixels) defined as *P*(*R*(*i*)). A threshold value representing a specific area threshold is denoted as *P*(*R*(*i*)). A threshold value representing a specific area threshold is denoted as *U*. The goal is to remove all regions *R*(*i*) from the set *S* whose area is not greater than *U*. This procedure can be mathematically described as follows:(4)$$W=Z\setminus \left\{R\left(i\right)\in S:P\left(R\left(i\right)\right)\left.\leq U\right\}\right.$$
where
*W*—the resulting binary image after removing smaller objects;*Z*—the original binary image;*R*(*i*)—the regions in the image *Z*;*S*—the set of all regions in the image *Z*;*P*(*R*(*i*))—the area of region *R*(*i*);*U*—the threshold value for the region’s area.

After image processing, lung segmentation is conducted to accurately identify lung regions necessary for analysis and diagnosis. The mask, which precisely defines the lung area, plays a crucial role in this process. The segmentation algorithm is directed only to this region, facilitating further processing and analysis [7,10]. This approach reduces processing time and enhances result accuracy (Figure 6).

The segmentation algorithm produces a binary image *X_mask_* of dimensions *M* × *N*, where *X_mask_*(*i*,*j*) = 1 denotes the lung region, and *X_mask_*(*i*,*j*) = 0 denotes the rest of the image. The mask is defined as *Q*(*i*,*j*), where *Q*(*i*,*j*)*ϵ*{0,1}, represents the region of interest. Applying the mask to the image is achieved through element-wise multiplication of the image *W* with the mask *Q*:(5)$$X_{mask}\left(i,j\right)=W\left(i,j\right)\cdot Q\left(i,j\right),$$
where
*X_mask_*(*i*,*j*)—the pixel value in the image after applying the mask at position (*i*,*j*);*W*(*i*,*j*)—the pixel value in the image *W* at position (*i*,*j*);*Q*(*i*,*j*)—the pixel value in the mask *Q* at position (*i*,*j*).

After successful segmentation, closing the segments becomes an important step. This operation employs morphological transformations to enhance the definition of lung regions, filling gaps and connecting partially interrupted contours. This results in better-defined regions that more accurately represent the lungs. The combination of these steps achieves high-quality results, which is crucial for further analysis, diagnostics, and research in medical and other fields where image processing is essential. The mathematical definition of the closing operation is as follows:(6)$$X_{closed}\left(i,j\right)=morphological\_closing\_function\left(X_{mask}\left(i,j\right),B_{closed}\left(i,j\right)\right)$$
where
*X_closed_*(*i*,*j*)—the pixel value at position (*i*,*j*) in the image resulting from the morphological closing operation applied to the segmentation image;*B_closed_*(*i*,*j*)—the structural element used in the closing process at pixel (*i*,*j*).

Visualization represents a crucial step in image processing, displaying the results of lung segmentation in a way that facilitates understanding and interpretation (Figure 6). This step involves presenting closed lung segments through a binary image clearly defined lung regions. Visualization is carried out by overlaying closed segments onto the original image through element-wise multiplication, specifically:(7)$$X_{visualization}\left(i,j\right)=X\left(i,j\right)\cdot X_{closed}\left(i,j\right)$$
where
*X_visualisation_*(*i*,*j*)—the pixel value at position (*i*,*j*) in the resulting image used for visualization;*X*(*i*,*j*)—the pixel value at position (*i*,*j*) in the original X-ray image.

By overlaying these segments on the original image, lung regions are emphasized with color in relation to the surrounding parts of the image, providing a visual representation relevant for further analysis [7,10].

### 2.4. Evaluation of Segmentation Performance

In the process of evaluating the performance of the proposed lung segmentation algorithm, key images used include the ground truth (manually segmented by experts) and the segmented image generated using our algorithm. To gain a comprehensive understanding, we analyzed the confusion matrix and used a set of key indicators, including specificity, sensitivity, accuracy, precision, F-measure, Matthew’s correlation coefficient (MCC), Dice index, Jaccard index, and area under the curve (AUC) analysis [6,15]. Each of these indicators provides specific information about the model’s capabilities, from sensitivity, which measures the detection of true positives, to AUC, which assesses the overall performance of the mode.

To calculate these performance indicators, we used elements of the confusion matrix, namely True Positive (*TP*), True Negative (*TN*), False Positive (*FP*), and False Negative (*FN*).

*Sensitivity* evaluates the model’s ability to correctly identify actual positive instances:(8)$$Sensitivity=\frac{TP}{\left(TP+FN\right)}$$

*Specificity* provides insights into the model’s ability to accurately identify the absence of changes:(9)$$Specificity=\frac{TN}{\left(TN+FP\right)}$$

*Accuracy* indicates the overall correctness of the model in labeling images, encompassing both accurate identifications and accurate absences of the condition of interest:(10)$$Accuracy=\frac{\left(TN+TP\right)}{\left(TP+TP+FP+FN\right)}$$

*Precision* measures the accuracy of the positive predictions:(11)$$Precision=\frac{TP}{\left(TP+FP\right)}$$

The *F*-measure, also known as the *F*1 score, is calculated as the harmonic mean between precision and recall (*sensitivity*):(12)$$F-measure=\frac{2\cdot Precision\cdot Sensitivity}{Precision+Sensitivity}$$

The *Dice* coefficient is used to measure the similarity between the ground truth and the segmented image. This coefficient provides information about the overlap between detected and actual damages in the image, and a value closer to 1 indicates greater similarity in segmentation. It is mathematically calculated according to the following formula:(13)$$Dice=\frac{2TP}{2TP+FP+FN}$$

*MCC* considers all four values from the confusion matrix, providing a measure of overall segmentation quality. It is calculated according to the following formula:(14)$$MCC=\frac{TP\cdot TN-FP\cdot FN}{\sqrt{\left(TP+FP\right)\left(TP+FN\right)\left(TN+FP\right)\left(TN+FN\right)}}$$

*Jaccard* Index, also known as IoU (Intersection over Union), measures the similarity between predicted and actual regions of interest and is calculated as the ratio of the intersection to the union of the two sets. The formula for the *Jaccard* Index is given using the following equation:(15)$$Jaccard=\frac{TP\cdot TN}{TP+FP+FN}$$

AUC measures the model’s ability to distinguish between the presence and absence of damage, providing a value between 0 and 1, where higher values indicate better performance.

These metrics together provide a comprehensive analysis, allowing us to gain a clear insight into the effectiveness of the algorithm in the context of medical lung segmentation.

## 3. Results

In this section, we will present the results of our research on newborns with RDS. To better understand the distribution of the data tracked in the study, Table 1 provides key values for the clinical parameters we monitored.

The average gestational week is 32.5, with a minimum value of 25 and a maximum of 36, indicating relative stability in the sample. The weight varies from 770 g to 3250 g, with an average of 2033.75 ± 623.03 g, suggesting wide variability. The fraction of inhaled oxygen before surfactant and inhaled corticosteroid treatment averages 53.97 ± 12.71%, decreasing to 20.09 ± 7.23% after treatment, indicating changes in oxygenation. The pH value before treatment is close to neutral, at 7.16, with a small variation of 0.11. The partial pressure of CO_2_ and HCO_3_ parameters shows appropriate average values and variations. BE before surfactant and inhaled corticosteroid treatment has an average value of −5.42 ± 5.76 mmol/L, while after treatment, it is 2.55 ± 2.91 mmol/L. Asymmetries of parameters range from −1.05 to 1.19, indicating different data distributions.

The examination of the effects of surfactant and inhaled corticosteroid on the respiratory parameters of newborns provided insights into significant changes in gas analyses during therapy. A paired sample *t*-test analysis was applied. Basic statistics related to pairs, as well as correlations, are presented in Table 2.

Before the administration of surfactant and inhaled corticosteroids, the average FiO_2_ value was 53.97 ± 12.71. After treatment, this value significantly decreased to 26.09 ± 7.23, indicating a reduction in oxygen levels after surfactant and inhaled corticosteroid therapy. The blood pH value before treatment (Before_pH) was 7.16 ± 0.11. After treatment (After_pH), the pH value increased to 7.39 ± 0.06. This increase indicates a change in blood acidity after the administration of surfactant and inhaled corticosteroids. In the analysis of the pCO_2_ parameter, the average CO_2_ level remained similar before and after treatment (Before_pCO2: 8.40 ± 1.52, After_pCO2: 5.93 ± 0.18), suggesting the preservation of CO_2_ after surfactant administration. The HCO_3_ parameter showed a slight change from 17.93 ± 4.73 before treatment to 26.49 ± 2.31 after treatment. The analysis of the BE parameter showed a significant increase in the average base value (Before_BE: −5.41 ± 5.76 to After_BE: 2.55 ± 2.91) after treatment with surfactant and inhaled corticosteroid, indicating a positive impact on the body’s acid-base balance. Although we observed correlations between certain parameters, they were not statistically significant.

Table 3 displays the differences between paired parameters with an assessment of the mean difference, standard deviation, standard error, 95% confidence interval, t-values, degrees of freedom, and *p*-values (two-tailed).

For the FiO_2_ parameter, a significant average difference of 27.88 ± 13.35 was observed between values before and after treatment with surfactant and inhaled corticosteroids, with a narrow 95% confidence interval. Analysis of blood pH values reveals a mean difference of −0.24 ± 0.12, also with a narrow confidence interval. Regarding the level of the pCO_2_ parameter, a difference of 2.47 ± 1.86 was noted, with a wider 95% confidence interval. For the pO_2_ parameter, a registered difference of −0.87 ± 2.52 was observed, with a wider confidence interval. The concentration of the HCO_3_ parameter shows an average difference of −8.56 ± 5.47, also with a wider confidence interval. For the BE parameter, the difference is −7.97 ± 6.55, with a wider confidence interval.

The parameters FiO_2_, pH, pCO_2_, HCO_3_, and BE show statistically significant changes, considering significance values less than 0.05. Although for the pO_2_ parameter, the significance value is not strictly less than 0.05 (Sig. = 0.061), this difference may indicate a potential change.

In order to analyze pathological changes in the lungs in more detail, we applied the developed algorithm for precise segmentation of lung tissue. The segmentation results for two characteristic cases, shown in Figure 7, provide a visual insight into characteristic changes during the X-ray image analysis process. It is important to emphasize that lung segmentation is performed depending on the degree of recovery, which can be identified in the X-ray image, thereby facilitating the definition of the Bomsell degree. This approach allows for a more precise analysis of pathology in terms of recovery, which plays a crucial role in understanding the evolution of lung changes in newborns with RDS.

In the first case depicted in Figure 7a, the initial X-ray image clearly falls into the V degree according to the Bomsell classification, evident through blurred lung tissues without a visible boundary of the cardiac shadow. In the second case shown in Figure 7b, the initial X-ray image clearly falls into the fourth degree according to the Bomsell classification, as evident through blurred lung tissue with air bronchograms over the cardiac shadow. It is important to note that the algorithm, in this specific case, does not perform image segmentation due to challenges caused by similar color throughout the X-ray image, resulting in unclear differentiation between soft tissues and bony structures. After the administration of a certain dose of surfactants, an improvement in the lung tissue condition is observed with the appearance of patchy-striped shadows above the segmented lung region and unclear boundaries of the heart. Although the presence of these shadows does not compromise the algorithm’s performance during segmentation, the overall lung structure becomes less visible due to the loss of a clear boundary of the heart. These characteristic structures in the X-ray image clearly indicate the III degree of lung damage according to the Bomsell classification while simultaneously signaling recovery in the segmented lung region. It is important to emphasize that the segmented parts of the lungs are incomplete, focusing on the outer region, where recovery is most pronounced. On the 16th day, the lungs are successfully depicted in their entirety, allowing the algorithm to perform uninterrupted segmentation. This complete representation covers all five global lung regions, enabling an overview of each region, including the upper, middle, lower, lateral, and mediastinal parts of the lungs. This comprehensive visualization provides a holistic insight into the lung’s condition, contributing to a thorough analysis of patient recovery.

Table 4 provides an overview of the performance evaluation indicator values for segmentation algorithms.

The segmentation results highlight a high specificity of 0.95, indicating precision in identifying negative instances. The algorithm’s sensitivity is 0.84, suggesting efficiency in recognizing positive instances. The overall segmentation accuracy reaches 0.93, providing a balance between precision and recall in predictions. The algorithm’s precision is 0.81, and the F-measure is 0.82, indicating a successful combination of accuracy and completeness in classification. The MCC rises to 0.78, while the Dice and Jaccard indices are 0.82 and 0.71, respectively, further confirming the algorithm’s effectiveness. An AUC value of 0.89 emphasizes a high level of performance, particularly in situations with imbalanced classes. These results serve as an indicator of the algorithm’s robustness in the segmentation process.

## 4. Discussion

In this study, which encompassed prematurely born infants treated with exogenous surfactant and inhaled corticosteroids, we analyzed the parameters of gas analysis in the arterial capillary blood of newborns before and after the implemented therapy. The results demonstrated that prior to the therapy, infants required higher fractions of inhaled oxygen. After the therapy, blood pH increased, the partial pressure of carbon dioxide decreased, and the partial pressure of oxygen increased. Additionally, a significant contribution of this study lies in the segmentation of radiographic chest images of newborns before, during, and after the conducted therapy.

A recently published study aimed at preventing bronchopulmonary dysplasia analyzed the impact of the instillation of inhaled corticosteroids with surfactant on inflammatory mediators in tracheal aspirate. In contrast to our study, the mentioned study did not show clinical benefits of the applied therapy but demonstrated an effective anti-inflammatory effect [20]. Unlike the previous study, a study conducted in Tubingen investigated the early use of inhaled corticosteroids in the prevention of bronchopulmonary dysplasia. The study included extremely premature infants (gestational age 23 weeks 0 days to 27 weeks 6 days), and unlike this study, the administration of inhaled corticosteroids continued until patients required oxygen therapy, respiratory support, or until reaching the 32nd postmenstrual week. This study demonstrated that the isolated inhalation of corticosteroids is considered responsible for reducing the incidence of bronchopulmonary dysplasia. Additionally, the study showed that the use of Budesonide had a favorable impact on the closure of the ductus arteriosus [13].

The research in this study clearly demonstrated that the application of surfactant in combination with inhaled corticosteroids shows statistical significance in almost all gas analysis parameters. Changes in FiO_2_, pH, pCO_2_, HCO_3_, and BE values showed statistically significant differences before and after the applied therapy, indicating a positive impact on respiratory function. After the administration of inhaled corticosteroids, the need for FiO2 was significantly reduced, a finding that was corroborated by Claus C. et al. in their research. Their study analyzed the impact of hydrocortisone administration on reducing respiratory support in prematurely born infants with bronchopulmonary dysplasia [21]. Similar results in blood gas analysis before therapy, as we obtained, were also shown by Moschino L et al. in their retrospective study analyzing the effects of two different types of therapy on RDS in extremely premature newborns [22,23].

Chest radiography is considered the “gold standard” for diagnosing respiratory distress syndrome. The routine use of chest radiographs prompted the objectification of this method, which we addressed in this study, all aimed at a more precise diagnosis of respiratory distress and the evaluation of applied therapy. In contrast to this research, Perri A. et al. recently analyzed the effectiveness of lung ultrasonography in prematurely born infants treated with surfactant. Similar to this study, images were analyzed at three time points: one before and two after surfactant therapy (before, 2 h after therapy, and 12 h after surfactant administration). They demonstrated that the use of lung ultrasonography can identify patients who will not require re-treatment with surfactant [24]. Debates on the superiority between lung ultrasonography and chest radiography remain current [25,26,27,28]. In one of the largest perinatal centers in Turkey, a prospective comparative analysis was conducted on the effectiveness of lung ultrasonography and chest radiography. The primary goal of this research was to predict the need for the next dose of surfactant and the therapeutic failure of continuous positive airway pressure applied [28]. As we have already mentioned, research is largely dedicated to RDS in newborns, considering that this syndrome represents a significant factor in neonatal mortality and morbidity. However, the complete confirmation of RDS in prematurely born infants still requires the implementation of radiological examinations, even though there is a growing trend towards the use of ultrasonography [5,19,29,30]. With the aim of expediting the diagnostic procedure and analyzing the degree of recovery in patients with RDS, we have also developed an algorithm for lung segmentation.

In general, research on lung region segmentation is often conducted in adults, applying various algorithmic approaches. Hamad Y.A. et al. investigated a similarity-based approach, focusing on the Otsu thresholding method, which facilitates the adjustment of the image into two distinct regions. This approach assumes that the image contains a dark object on a light background or vice versa, and in lung segmentation on X-ray images, the dark area represents the lungs, while other parts represent the background [31].

In the development of our approach, we found inspiration in the work of Mehmood M. et al. [9], who successfully applied a similar methodology for breast cancer segmentation and detection. Similarly, the work of Vijh S. et al. [16] further influenced our approach, confirming the effectiveness of threshold-based segmentation for extracting lung segments, along with the correction of imperfections using morphological operations and successful lung cancer detection. This work represents a synthesis of these inspirations, applying them to our research approach to achieve precise segmentation of the lung region and enhance the understanding of medical diagnostic images in prematurely born infants with RDS. Highlighting that this algorithm has significantly contributed to advancements in research through carefully tailored lung region segmentation methods is essential. The combination of a threshold-based approach and morphological operations has resulted in high precision in lung region segmentation. The success achieved affirms the importance of this research in the context of medical diagnostics in neonatology. Simultaneously, the obtained results validate the applicability of our approach in a broader scientific context [7,10,19]. It is common practice to analyze confusion matrix indicators to evaluate the performance of developed algorithms [6,12] providing insights. A value of 0.82 represents a significant step towards accurate segmentation in terms of its effectiveness and validity in the context of segmenting lung regions in newborns with RDS. Our algorithm stands out by achieving exceptional results, making a significant contribution to the field of medical lung region segmentation in newborns. With a specificity of 0.95, sensitivity of 0.84, global accuracy of 0.93, precision of 0.81, F-measure of 0.82, Matthew’s correlation coefficient of 0.78, and Dice and Jaccard indices of 0.82 and 0.71, it attests to the consistency and accuracy of our algorithm.

The AUC value of 0.89 further emphasizes a high level of performance, especially in situations with imbalanced classes, confirming the significance and relevance of this work in medical research, with a focus on lung region segmentation in newborns with RDS. While other researchers have achieved similar results in terms of analyzing the performance of their algorithms [7,9,15], our algorithm stands out as a significant contribution to medical diagnostics in this sensitive population segment. These results contribute to a broader understanding and improvement of the field of medical segmentation in newborns with RDS.

This study focuses on classical lung segmentation methods, standing out through an approach based on traditional techniques. Direct comparison with widely used deep learning techniques becomes challenging due to fundamental differences in methodologies. However, analyzing algorithm performance through parameters like Dice coefficient, AUC, sensitivity, and accuracy provides intriguing insights. Training deep learning algorithms often involves extensive datasets, leading to improved performances. For instance, Kim M. and Lee B.-D., as well as Arsalan M. et al., achieved impressive results in deep learning algorithms, especially in terms of Dice coefficient and sensitivity [32,33], which, compared to ours, are slightly higher. It is important to note that all the studies mentioned were focused on region segmentation in the lung and chest area, creating a common framework for comparison. On the other hand, Khoiriyah S.A. et al. and Zhang J. et al. achieved significantly lower results in terms of AUC and sensitivity in lung segmentation despite having extensive data in their databases [34,35], while Siddiqi R achieved results similar to ours [36].

The results of this study provide significant insights into the effectiveness of applied therapeutic approaches in newborns with RDS. It is important to note that, although these results are valuable, the research is not without certain limitations. A small number of patients were carefully selected, contributing to high sample homogeneity but simultaneously limiting the overall generalization of results. Ethical aspects of neonatal research, especially considering the vulnerable population of premature infants, require a cautious approach to patient recruitment. Short-term monitoring of therapy effects represents an additional limitation, complicating a full understanding of long-term outcomes and potential complications. This fact emphasizes the necessity of future research covering an extended time period and conducting a more detailed analysis of the long-term effects of applied therapeutic strategies.

Integrating MATLAB algorithms directly into clinical practice through the development of an intuitive Graphical User Interface (GUI) will facilitate physicians’ seamless use of algorithms in real time, enhancing the diagnostic process. This implementation aims to increase the accessibility and practical application of our algorithm in real clinical settings, enabling physicians to have quick and efficient access to relevant information from medical images. The use of deep machine learning and MATLAB R2012b GUIs can significantly speed up the diagnostic process and has found wide application in clinical practice. This software enables doctors to effectively communicate with data, analyze it, and make decisions through an intuitive user interface [37,38,39].

## 5. Conclusions

In the presented research, we analyzed the effectiveness of applied therapeutic interventions, exogenous surfactants, and inhaled corticosteroids in prematurely born infants with RDS. Our analysis of gas parameters in arterial capillary blood showed significant improvements in respiratory functions after the implemented therapies. Simultaneously, we developed and applied an innovative algorithm for lung segmentation in radiographic images of newborns, providing additional value to RDS diagnosis. The combination of a threshold-based approach and sophisticated morphological operations allowed our algorithm to extract lung regions consistently and accurately, emphasizing its crucial role in medical segmentation.

Our research not only confirms the crucial role of applied therapies but also highlights the importance of the developed algorithm in the field of medical research, particularly in enhancing lung segmentation in the sensitive population of newborns with RDS. We continue further research with a focus on improving the proposed algorithm through the development of modules for the classification of radiographic images, aiming for continuous enhancement of diagnostic methods in neonatology.

## Figures and Tables

**Figure 1 diagnostics-14-00214-f001:**
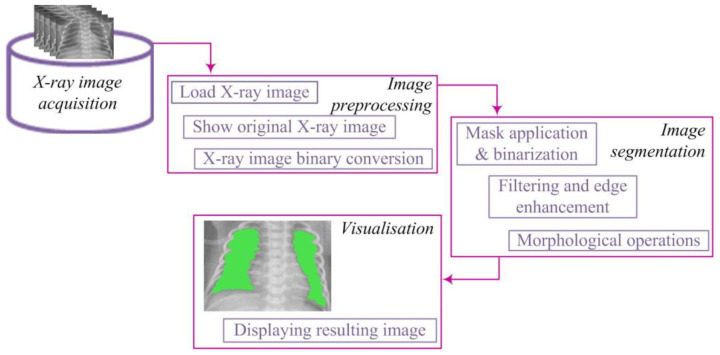
Lung segmentation algorithm.

**Figure 2 diagnostics-14-00214-f002:**
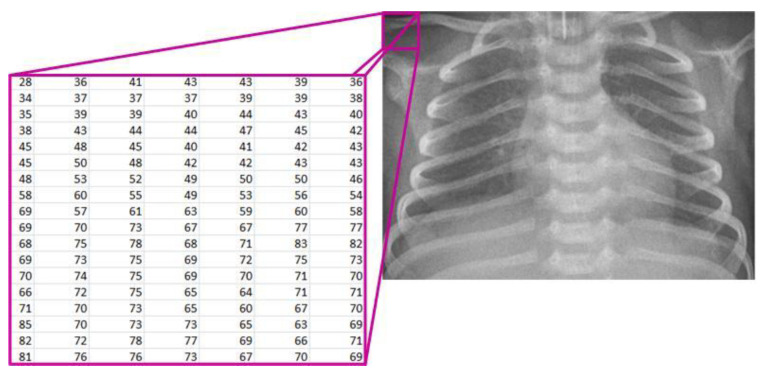
Mathematical representation of the X-ray image.

**Figure 3 diagnostics-14-00214-f003:**
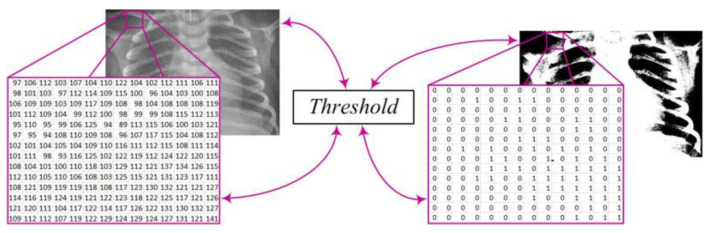
X-ray image binary conversion.

**Figure 4 diagnostics-14-00214-f004:**
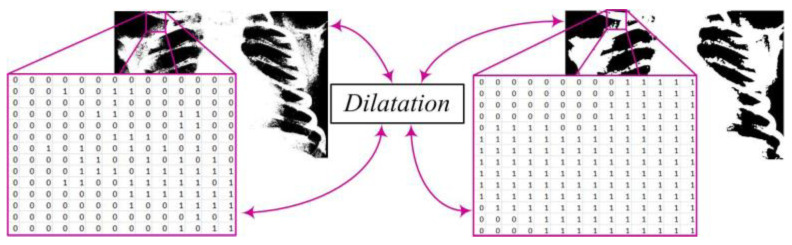
Enhancing object boundaries through image dilatation.

**Figure 5 diagnostics-14-00214-f005:**
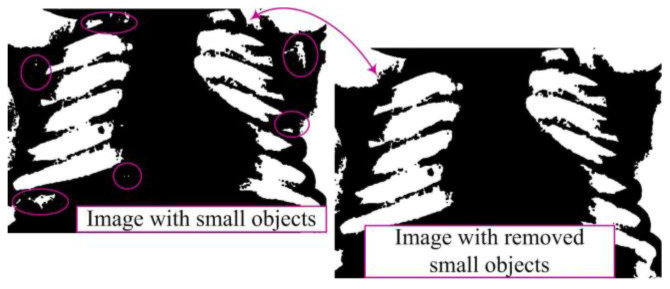
Removal of small objects in binary image.

**Figure 6 diagnostics-14-00214-f006:**
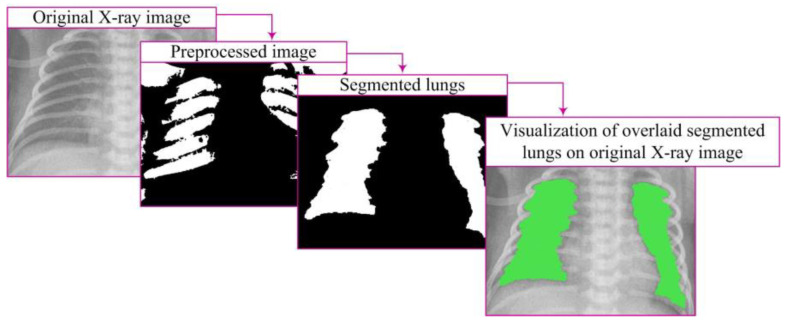
Segmentation process and visualization.

**Figure 7 diagnostics-14-00214-f007:**
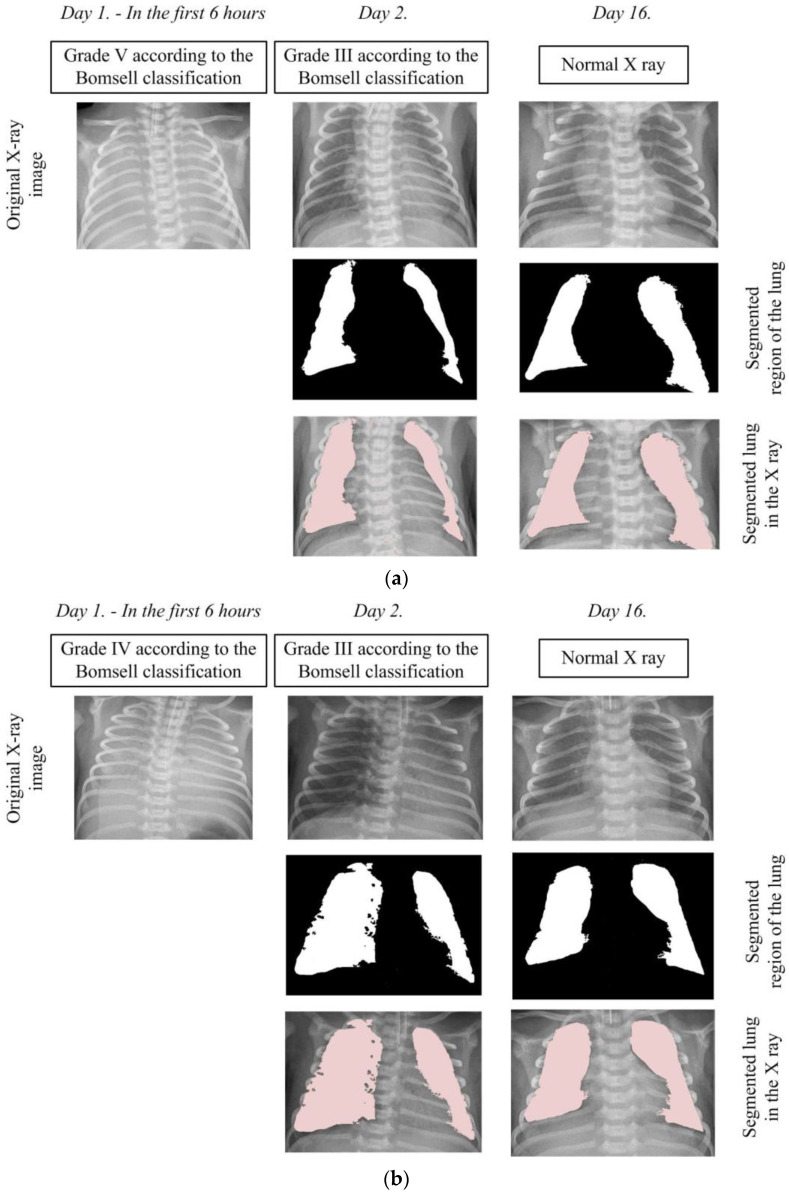
Segmentation of the lung in the X-ray images: (**a**) Case 1 and (**b**) Case 2.

**Table 1 diagnostics-14-00214-t001:** Clinical parameters in neonates with RDS.

Parameter	Range	Min	Max	Mean ± S.D.	Skewness ± S.D.
Gestational week	11	25	36	32.5 ± 2.95	−1.05 ± 0.42
Weight, g	2480	770	3250	2033.75 ± 623.03	−0.21 ± 0.42
Before_FiO_2_, %	49	40	89	53.97 ± 12.71	1.19 ± 0.42
After_FiO_2_, %	19	21	40	20.09 ± 7.23	1.05 ± 0.42
Before_pH,/	0.42	6.90	7.32	7.16 ± 0.11	−0.59 ± 0.42
After_pH,/	0.26	7.25	7.51	7.39 ± 0.06	−0.17 ± 0.42
Before_pCO_2_, kPa	6.20	5.80	12	8.40 ± 1.52	0.47 ± 0.42
After_pCO_2_, kPa	5.20	3.20	8.40	5.93 ± 1.03	−0.33 ± 0.42
Before_pO_2_, kPa	10.20	2.30	12.50	5.79 ± 2.38	0.65 ± 0.42
After_pO_2_, kPa	5.10	4.30	9.40	6.66 ± 1.24	0.37 ± 0.42
Before_HCO_3_, mmol/L	26.10	2.20	28.30	17.93 ± 4.73	−1.07 ± 0.42
After_HCO_3_, mmol/L	11.20	22.10	33.30	26.29 ± 2.37	0.6 ± 0.42
Before_BE, mmol/L	31.60	−16.70	14.90	−5.42 ± 5.76	1.07 ± 0.42
After_BE, mmol/L	12.20	−3.10	9.10	2.55 ± 2.91	0.15 ± 0.42

Legend: FiO_2_—oxygen fraction; pCO_2_—partial pressure of carbon dioxide; pO_2_—partial pressure of oxygen; HCO_3_—bicarbonate concentration; BE—base excess; and S.D.—standard deviation.

**Table 2 diagnostics-14-00214-t002:** Paired sample *t*-test statistics and correlations.

Paired Parameters		Mean ± S.D	Correlation	Sig.
FiO_2_	Before_FiO_2_	53.97 ± 12.71	0.19	0.29
	After_FiO_2_	26.09 ± 7.23		
pH	Before_pH	7.16 ± 0.11	−0.02	0.94
	After_pH	7.39 ± 0.06		
pCO_2_	Before_pCO_2_	8.40 ± 1.52	−0.04	0.81
	After_pCO_2_	5.93 ± 0.18		
pO_2_	Before_pO_2_	5.79 ± 2.38	0.15	0.42
	After_pO_2_	6.66 ± 1.24		
HCO_3_	Before_HCO_3_	17.93 ± 4.73	−0.11	0.59
	After_HCO_3_	26.49 ± 2.31		
BE	Before_BE	−5.41 ± 5.76	−0.04	0.85
	After_BE	2.55 ± 2.91		

Legend: FiO_2_—oxygen fraction; pCO_2_—partial pressure of carbon dioxide; pO_2_—partial pressure of oxygen; HCO_3_—bicarbonate concentration; BE—base excess; S.D.—standard deviation; and Sig.—significance.

**Table 3 diagnostics-14-00214-t003:** Paired differences.

Paired Parameters	Mean ± S.D	Std. Err.	95% Conf. Int. of the Diff.		t	df	Sig. (2-Tailed)
			Lower	Upper			
FiO_2_	27.88 ± 13.35	2.36	23.06	32.69	11.82	31	0.000
pH	−0.24 ± 0.12	0.02	−0.29	−0.2	−11.35	31	0.000
pCO_2_	2.47 ± 1.86	0.33	1.79	3.14	7.45	31	0.000
pO_2_	−0.87 ± 2.52	0.45	−1.77	0.04	−1.95	31	0.061
HCO_3_	−8.56 ± 5.47	0.99	−10.61	−6.52	−8.57	29	0.000
BE	−7.97 ± 6.55	1.16	−10.33	−5.61	−6.88	31	0.000

Legend: FiO_2_—oxygen fraction; pCO_2_—partial pressure of carbon dioxide; pO_2_—partial pressure of oxygen; HCO_3_—bicarbonate concentration; BE—base excess; S.D.—standard deviation; and Sig.—significance.

**Table 4 diagnostics-14-00214-t004:** Values of performance evaluation indicators for segmentation algorithms.

Indicator	Value ± S.D.
Specificity	0.95 ± 0.05
Sensitivity	0.84 ± 0.16
Accuracy	0.93 ± 0.06
Precision	0.81 ± 0.16
F-measure	0.82 ± 0.14
MCC	0.78 ± 0.15
Dice	0.82 ± 0.14
Jaccard	0.71 ± 0.21
AUC	0.89 ± 0.09

S.D—standard deviation.

## Data Availability

The datasets used and analyzed in the current study have been made available from the corresponding author on reasonable request.

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
