# Peer review of "Advanced Diagnostics of Respiratory Distress Syndrome in Premature Infants Treated with Surfactant and Budesonide through Computer-Assisted Chest X-ray Analysis"

_diagnostics, 2024, doi:10.3390/diagnostics14020214_

Round 1

Reviewer 1 Report

Comments and Suggestions for Authors

I examined your study titled "Advance Diagnostic of Respiratory Distress Syndrome in Premature Infants Treated with Surfactant and Budesonide through Computer-Assisted Chest X-ray Analysis" in detail. I want to point out that the study is written well in general terms. However, there are some shortcomings in the study. I write down the deficiencies I see in items. No literature review was conducted. You can review the related study "Classification of computerized tomography images to diagnose non-small cell lung cancer using a hybrid model", which was done using chest X-ray images. What methods were used for the image visualization process? This part has remained very close. Similarly, the methods used for the segmentation process should also be detailed. Especially adding a short and understandable subparagraph in the form of the proposed model in the article I mentioned will increase the article's readability. Finally, the limitations of the study should be included. Words like our, we are used a lot in the article. These words should be avoided as much as possible.

Comments on the Quality of English Language

Spelling and grammatical errors should be reviewed.

Reviewer 2 Report

Comments and Suggestions for Authors

Dear authors,

I have now completed the review of the manuscript titled "Advance Diagnostic of Respiratory Distress Syndrome in Premature Infants Treated with Surfactant and Budesonide through Computer-Assisted Chest X-ray Analysis"

In the present study, the authors explored the use of algorithms in diagnosing and monitoring Respiratory Distress Syndrome (RDS) in preterm newborns, with a specific focus on the application of surfactants and inhalation corticosteroid therapy. The research demonstrates significant findings and contributions to the field of neonatal care and medical imaging. 

The manuscript is interesting and, in general, fair written.

However, some critical evaluations and constructive criticisms could be considered. I would like to suggest that the authors address these limitations in the article, either by discussing them in the limitations section or, where feasible, by making the appropriate revisions:

1. The study included 32 preterm newborns, which may be considered limited for representing the broad spectrum of RDS severity and responses to treatment. A larger, more diverse cohort could enhance the validity and generalizability of the findings.

2. While the segmentation algorithm achieved promising results with high global accuracy, precision, and F-score, it is crucial to validate these findings against other established methods or across different institutions with varied settings. Further external validation with a larger dataset might strengthen the confidence in the algorithm's performance.

3. The research is positioned in the context of the current literature, yet it would benefit from a more detailed comparison with other segmentation techniques, particularly deep learning approaches that have shown significant results in medical image analysis.

4. The study focuses on the immediate diagnostic and treatment phase of RDS. Expanding the research to include long-term outcomes and how early diagnosis and treatment alterations impact the prognosis and quality of life of these infants would provide a more comprehensive understanding of the clinical implications of the study.

5. The study utilizes MATLAB for algorithm implementation and image analysis. Discussing the potential for integration into clinical workflows, ease of use for healthcare professionals, and scalability across different hospital systems would be beneficial. Addressing how this technology can be adapted or implemented in resource-limited settings would also be valuable.

Thank you for your valuable contributions to our field of research. I will wait a revised manuscript.

Round 2

Reviewer 1 Report

Comments and Suggestions for Authors

Thank you for addressing the shortcomings I pointed out.

Comments on the Quality of English Language

.